# Bridging Classical Methodologies in *Salmonella* Investigation with Modern Technologies: A Comprehensive Review

**DOI:** 10.3390/microorganisms12112249

**Published:** 2024-11-07

**Authors:** Steven Ray Kitchens, Chengming Wang, Stuart B. Price

**Affiliations:** Department of Pathobiology, College of Veterinary Medicine, Auburn University, 1130 Wire Road, Auburn, AL 36849-5519, USA; srk0002@auburn.edu (S.R.K.); wangche@auburn.edu (C.W.)

**Keywords:** *Salmonella*, virulence, environment, bacteriophage, epidemiology, computational biology

## Abstract

Advancements in genomics and machine learning have significantly enhanced the study of *Salmonella* epidemiology. Whole-genome sequencing has revolutionized bacterial genomics, allowing for detailed analysis of genetic variation and aiding in outbreak investigations and source tracking. Short-read sequencing technologies, such as those provided by Illumina, have been instrumental in generating draft genomes that facilitate serotyping and the detection of antimicrobial resistance. Long-read sequencing technologies, including those from Pacific Biosciences and Oxford Nanopore Technologies, offer the potential for more complete genome assemblies and better insights into genetic diversity. In addition to these sequencing approaches, machine learning techniques like decision trees and random forests provide powerful tools for pattern recognition and predictive modeling. Importantly, the study of bacteriophages, which interact with *Salmonella*, offers additional layers of understanding. Phages can impact *Salmonella* population dynamics and evolution, and their integration into *Salmonella* genomics research holds promise for novel insights into pathogen control and epidemiology. This review revisits the history of *Salmonella* and its pathogenesis and highlights the integration of these modern methodologies in advancing our understanding of *Salmonella*.

## 1. Introduction

*Salmonella enterica* subspecies *enterica* (commonly referred to as *Salmonella*), specifically, non-typhoidal *Salmonella*, is a major concern in foodborne-associated illness in humans in the United States of America (USA) [1]. According to the recently released USA Centers for Disease Control and Prevention (CDC) Laboratory-based Enteric Disease Surveillance (LEDS) system, *National Enteric Disease Surveillance:* Salmonella *Annual Report, 2016*, typhoidal and paratyphoidal serovars of *Salmonella* accounted for 941 human cases out of 47,424 total culture-confirmed *Salmonella* infections [2]. On a global scale, in 2017, typhoid and paratyphoid disease accounted for an estimated 14.3 million human cases [3]. Global estimations of non-typhoidal *Salmonella* gastroenteritis are 93.8 million human cases [4]. On a global scale, typhoidal and paratyphoidal salmonellosis is a larger concern [3,4], but non-typhoidal salmonellosis is a larger concern in the USA [2]. This review will primarily focus on non-typhoidal *Salmonella* infections and thus epidemiological data from the USA were selected for review. Non-typhoidal *Salmonella* disease in humans is primarily foodborne and associated with food-producing animal sources, so this review will primarily focus on *Salmonella* in animals [5]. This review examines *Salmonella* from a historical perspective in light of current technologies and approaches.

The genus *Salmonella* is a Gram-negative facultatively anaerobic and peritrichously flagellated bacilli. *Salmonella* can be distinguished from members of other genera of the *Enterobacteriaceae* by a combination of biochemical reactions such as the production of hydrogen sulfide, citrate metabolism, lysine as a nitrogen source, and tetrathionate as a terminal electron acceptor [6]. It was named after the American veterinarian Daniel E. Salmon, who first isolated *Bacillus cholera*-suis from a pig suffering from hog cholera [7,8]. *B. cholera*-suis has since been renamed *Salmonella enterica* subspecies *enterica* serovar Choleraesuis, abbreviated to *Sal*. *enterica* serovar Choleraesuis or *Sal*. Choleraesuis.

The history of the taxonomy of *Salmonella* species is complicated and controversial with phenotypic, serologic, and genotypic methods used to determine the phylogeny of *Salmonella* [9,10]. Originally, each serovar of *Salmonella* was classified as a separate species. *Salmonella* is now comprised of two species, *Salmonella enterica* and *Salmonella bongori*. *Salmonella enterica* contains six subspecies (*enterica, indica, salamae, houtenae, diarizonae,* and *arizonae*), and recent genome-based studies have proposed an additional eleven subspecies [11,12,13,14]. *Salmonella enterica* subspecies *enterica* comprises over 2600 serovars [11,12]. As an example of the nomenclature for *Salmonella* (*Sal*.) *enterica* subspecies *enterica* serovars, the *Salmonella enterica* subspecies *enterica* serovar Typhimurium can be simplified with the synonyms *Sal*. *enterica* sub. *enterica* Typhimurium or simply *Sal*. Typhimurium. These serovars are designated by the antigenic formula, which incorporates the antigenic properties of their lipopolysaccharide (LPS) sugar repeat units (O-antigens) and their flagellar structural protein subunits (H-antigens). In a few serotypes, *Sal*. Typhi, *Sal*. Dublin, and *Sal*. Paratyphi C, a capsular polysaccharide antigen (Vi-antigen) can be found. The method of using slide agglutinations for Vi- and O-antigens and slide or tube agglutinations for H-antigens results in formulae that can be deciphered to determine the name of a serovar using the White–Kauffmann–Le Minor (WKLM) scheme [9,15]. As an example, the antigenic formula for *Salmonella* Newport is 6,8,20:e,h:1,2:[z67],[z78] and the antigenic formula for *Salmonella* Typhimurium is 1,4,[5], 12:i:1,2. These antigenic formulas can be broken down with the example of *Salmonella* Newport where the 6, 8, and 20 are the O-antigens, the e,h represents the phase 1 H-antigens, the 1,2 represents the phase 2 H-antigens, and [z67],[z78] are considered special antigens. *Salmonella* Typhimurium antigenic formula is broken down to where the 1, 4, [5], and 12 are the O-antigens, the i represents the phase 1 H-antigen, and the 1,2 represents the phase 2 H-antigens.

The standard serotyping method involves using rabbit antisera with antibodies specific to the individual antigens that comprise the WKLM scheme. The WKLM scheme has not been updated since 2007, and there is no consensus on how to replace it [16]. Serologic *Salmonella* serotyping reagents are expensive, laboratories are highly specialized, the method is laborious and time-consuming, it requires well-trained technicians, and results can be open to interpretation error [17,18]. Researchers at the United States of America (USA) Centers for Disease Control and Prevention (CDC) started sequencing the alleles on the genes that encode the flagella, *fliB*, *fliC*, and *flpA* [19]. From this work, a deoxyribonucleic acid (DNA) bead-based liquid array was used, and specific polymerase chain reaction (PCR) primers were designed for genes encoding the flagellar antigens (*fliB* and *fliC*) and probes for the determination of 36 flagellar antigen genes of *Salmonella* [20]. Also, at the CDC, Fitzgerald et al. (2007) developed a related strategy for serogroup identification based on the O-antigen *rfb* genes, from which signature probes were derived and integrated into a suspension bead (Luminex Technology) fluorescence assay [21].

*Salmonella* serotyping is in transition. *Salmonella* serotyping still uses classical methods such as slide and tube agglutinations for detection of the Vi-, O-, and H-antigens, but genomic typing tools have become increasingly popular with the rise of next-generation sequencing (NGS) techniques [16,22]. In 1988, MultiLocus Enzyme Electrophoresis (MLEE) was used to identify natural groupings of *Salmonella*. These groupings could correspond to serovars, and a sequence-based alternative, MultiLocus Sequence Typing (MLST), was developed [23]. MLST was like MLEE but was based on sequences of multiple housekeeping gene fragments as opposed to electrophoretic migration of proteins [23]. Zhang et al. (2015), developed a sequence-based method of serotyping (“SeqSero”) that incorporated a curated database that included *rfb* gene clusters responsible for somatic O-antigen synthesis; *wzx* O-antigen flippase gene; *wzy* O-antigen polymerase gene; additional genes from the *rfb* cluster that is useful for O-group determination; sequence-specific genetic markers for additional O-antigen groups; and the *fliC* and *fliB* genes that encode *Salmonella* flagellar antigens. Raw sequence reads are mapped against the curated database, or with genome assemblies, genes of interest are extracted and mapped against the curated database [24]. Yoshida et al. (2016) developed a core gene MLST (cgMLST) method called *Salmonella* In Silico Typing Resource (SISTER). Yoshida et al. characterized the method as a genoserotyping approach that incorporates queried genome assemblies into cgMLST-based phylogenetic clusters [25]. Zhang et al. (2019) developed an updated tool (“SeqSero2”) with an expanded database to serotype raw sequence reads or assemblies. SeqSero2 generates k-mers from assemblies or Oxford Nanopore Technology (ONT) reads. The query genome’s O- or H-antigen genes were matched to a database that yielded the highest similarity score. With raw sequence reads, micro-assemblies were generated and mapped to the curated database [26]. SISTR and SeqSero2 are well-recognized bioinformatic tools, and the original SeqSero is widely used as it is accessible through the Center for Genomic Epidemiology (https://cge.food.dtu.dk/services/SeqSero/ (accessed on 19 September 2024)) and on the BioNumerics software platform (https://www.applied-maths.com/bionumerics (accessed on 19 September 2024)) [26,27].

## 2. Salmonellosis and the Host Specificity

The nomenclature of *Salmonella* is very complex, but the classification of *Salmonella* serovars that differ in the host range of clinical salmonellosis is also very complicated. *Salmonella* serovars are genetically closely related, but there are wide variations in host-specificity, virulence, and disease manifestations.

Salmonellosis can manifest as a range of symptoms from the asymptomatic carrier stage, enterocolitis/diarrhea, to the life-threatening bacteremia/septicemia and enteric/typhoid fever [28,29,30,31,32]. *Salmonella* can also be categorized as typhoidal and non-typhoidal *Salmonella*, which is whether the *Salmonella* serovar manifests a systemic typhoid-like fever or a more common, self-limiting gastroenteritis (non-typhoidal disease) that accounts for the foodborne illness typically seen in the USA [33,34]. Typhoidal serovars are serovars such as *Sal.* Typhi, *Sal*. Paratyphi A, *Sal*. Paratyphi B, *Sal*. Paratyphi C, and *Sal*. Sendai only infects humans and higher primates [35,36]. Several serovars cause typhoid-like bacteremia in specific animal hosts such as *Sal*. Cholerasuis in pigs, *Sal*. Dublin in cattle, *Sal*. Typhimurium in mice, *Sal*. Gallinarium in poultry, *Sal.* Pullorum in poultry, and *Sal.* Abortusovis in sheep. Some typhoid-like serovars are considered non-typhoidal serovars in different animal hosts, such as *Sal*. Typhimurium causes a typhoid-like fever in mice but causes gastroenteritis in humans, cattle, and horses [30,36,37,38].

*Salmonella* is also divided into groups based on their host range: “non-adapted” (broad), “host-adapted”, and “host-restricted” [31,39,40,41]. Host-specific *Salmonella* has a very narrow host range (usually one specific host species) and causes typhoid or typhoid-like disease, with examples that include *Sal.* Typhi (humans and higher primates), *Sal*. Gallinarum (poultry), *Sal*. Abortusovis (sheep), *Sal*. Typhisuis (pigs), and *Sal*. Abortusequi (horses) [30,39,40,42]. Host-adapted *Salmonella* has a narrow host range with the ability to disseminate beyond the gastrointestinal tract, colonize systemic sites, persist systemically for long periods, possibly persistent asymptomatic infections, and often are vertically transmitted in their preferred hosts [30,31,41,43]. *Sal*. Dublin (cattle) and *Sal*. Cholerasuis (pigs) can cause systemic disease and bacteremia and be vertically transmitted in their preferred host but can accidentally infect other species, such as humans [30,31]. The non-preferred host usually exhibits subclinical infections [40]. Some infections in non-preferred hosts might be localized to unusual locations, such as human cases involving a chest wall abscess (*Sal*. Cholerasuis) and a thyroid abscess (*Sal*. Dublin) [44,45].

Broad-host range *Salmonella* are serovars that infect and cause disease in a wide range of host species and exhibit what is characterized as non-typhoidal disease. This type of disease is self-limiting, with acute gastroenteritis, and watery diarrhea. The host’s inflammatory response is responsible for the symptoms of diarrhea, nausea, vomiting, intestinal cramping, and fever [6,46]. Gastroenteritis found in humans can also be seen in infected animals. Subclinical infections in animals are common. Subclinical symptoms can be reduced milk or egg production, reduced weight gain, and persistent carrier states [47]. *Salmonella* in humans is primarily a foodborne pathogen associated with food-producing animal sources [5]. Animals can be infected by close contact with infected animals, contaminated water or direct contact with feces or feces-contaminated equipment, contaminated feed or environment, or potential transmission by arthropods [47]. Recirculation of *Salmonella* in the environment can lead to animals being reinfected and the appearance of a persistent carrier state in animals [6].

## 3. *Salmonella* Virulence Factors Associated with Gastroenteritis

### 3.1. Salmonella Pathogenicity Islands

*Salmonella* Pathogenicity Islands (SPIs) are clusters of virulence genes found on the chromosome. These SPIs encode factors essential for adhesion, invasion, survival, and replication within a host [48]. There are twenty-four known SPIs with SPI-1, SPI-2, SPI-3, SPI-4, SPI-5, SPI-6, SPI-9, and SPI-11 being conserved across all *Salmonella* [49].

The SPI-1 encodes the type III secretion system (T3SS-1), regulators, effector proteins, and chaperone proteins [50]. Type III secretion systems (T3SS) are complex membrane molecular machines also called injectisomes. It injects bacterial effector proteins into a eukaryotic host cell [51]. SPI-1, T3SS-1, and T3SS-1 effector proteins are essential for host cytoskeleton rearrangement and invasion of epithelial cells [48]. The T3SS-1 effector proteins are encoded on SPI-1 (SipA, SipB, SipC, SipD, SptP) and SPI-5 (SopB) [6].

The SPI-2 and its encoded T3SS-2 are a crucial virulence factor required for macrophage survival [6]. *Salmonella* will be contained within the *Salmonella*-containing vacuole (SCV) in infected host cells such as epithelial cells and macrophages. The T3SS-2 is responsible for injecting effector proteins across the membrane of the SCV [52]. Twenty-eight known effector proteins can be translocated by the T3SS-2, with only a few encoded on the SPI-2 [48]. SPI-2 is essential for virulence in *Salmonella*. Grant et al. (2012) found that *Salmonella* with a mutation in the SPI-2 T3SS-2 could not replicate inside or escape from an infected cell [53].

The SPI-1 T3SS effectors trigger the production of proinflammatory cytokines and the less well-characterized SPI-2 T3SS-2 proinflammatory activity, which stimulates the rapid recruitment of neutrophils and induces acute intestinal inflammation and gastroenteritis [52,54]. This response is exacerbated by SPI-1-dependent induction of macrophage cell death [52]. The host cell death is induced by SPI-1 effectors as well as SPI-2 effectors; this results in programmed cell death and further dissemination of Salmonellae [55]. Some T3SS-secreted effectors have the potential to reduce inflammatory responses and halt over-activated innate immune responses, which may help avoid detrimental endpoints for the host upon infection. The T3SSs of *Salmonella* are very complex in their ability to provoke strong inflammatory responses and suppress the inflammatory response to provide the best environment for *Salmonella* to replicate [56].

### 3.2. Surface Structure—LPS and the “O”-Antigen

LPS is a molecule associated with Gram-negative bacteria. It is an outer membrane (OM) molecule comprised of three structural regions: the hydrophobic region called lipid A (or endotoxin), the nonrepeating core oligosaccharide, and the distal O side-chain polysaccharide (or the O-antigen). LPS is anchored to the OM by lipid A. The core oligosaccharide is highly conserved among *Enterobacteriaceae* and is an attachment site for the variable O-antigen. It is encoded on the *rfb* gene cluster [57]. *Salmonella* colonies with the full O side chain have a smooth appearance and are referred to as smooth. While *Salmonella* mutant colonies that have lost their O side chain have dull surfaces and are referred to as rough mutants.

LPS establishes a permeable barrier that protects the cell from toxic molecules such as antibiotics and bile salts. LPS is the primary bacterial component encountered by the host immune system. Toll-like receptor 4 (TLR4) recognizes it and binds to lipid A, which activates the expression of pro-inflammatory cytokine genes and apoptosis [57]. The core oligosaccharide is important for serum, antimicrobial peptide, and bile salt resistance [58].

The O-antigen is encoded on the *rfb* gene cluster. The O-antigen side chain is the outermost portion of the LPS. This portion of the LPS is the O-antigen used for serovar identification by the WKLM scheme. Many antigenic factors exist as there are 46 serological-specific O-antigens used to represent the 46 recognized serogroups [10]. The O-antigen repeat unit contains three sugars and is present in variable numbers of repeat units ranging up to 40 units [59]. The O-antigen is critical in resistance to complement-mediated lysis with true rough isolates with no O-antigen side chain being much more sensitive to serum killing [48,60]. At the same time, complement can bind to the O-antigen, where complement C3 is important for marking the cell for phagocytosis and complement C5 is an important proinflammatory chemoattractant [61]. Modification of the LPS structures can impair complement recognition and is a common resistance mechanism of Gram-negative bacteria [60]. It has been shown that mutations that produce a truncated O-antigen might have resistance to complement. Murray et al. found that O-antigens greater than fifteen sugar repeat units were necessary for complement activation and less than four sugar repeat units for complement-mediated lysis [62].

### 3.3. Surface Structure—Flagella the “H”-Antigen

The Flagellar protein, also known as the “H”-antigen, is the motility structure for *Salmonella*. *Salmonella* has two distinct H-antigens, phase 1 (H_1_-antigen) and phase 2 (H_2_-antigen). These are encoded on the *fliC* gene (H_1_-antigen) and *fljB* gene (H_2_-antigen). *Salmonella* can alternately express between the two flagellar genes known as phase variation [19,63]. This ability to change its flagellar proteins helps it avoid being cleared by the host’s immune system [64]. Serovars that can express both H_1_-antigen(s) and H_2_-antigen(s) are called diphasic, while serovars that only express the H_1_-antigen(s) are called monophasic [19,63].

### 3.4. Surface Structure—The Capsular or “Vi”-Antigen

The virulence or “Vi”-antigen is a polysaccharide capsule encoded on SPI-7 [65,66,67]. SPI-7 is one of *Salmonella*’s largest excisable pathogenicity islands but is only found in three serovars (*Sal*. Typhi, *Sal*. Paratyphi C, and *Sal*. Dublin) [68]. The main difference between the SPI-7 of *Sal*. Typhi versus *Sal*. Paratyphi C, and *Sal*. Dublin is that *Sal*. Typhi SPI-7 encodes a T3SS-1 effector protein SopE [48]. The Vi-antigen capsule is thought to enhance systemic virulence by increasing bacterial resistance to complement, reducing phagocytic killing by protecting the bacterium from reactive oxygen species (ROS), and interfering with pathogen-associated molecular pattern (PAMP) activation of the innate immune system [65]. Vi-antigen expression represses T3SS-1 and flagella [69].

### 3.5. Salmonella and the Microbiota

The gut microbiota relies on fermentation to produce energy for growth. Epithelial cells detoxify microbiota-derived hydrogen sulfide (H_2_S) by conversion into thiosulfate (Figure 1) [70]. Microbes depend on the nutrients present in the mucous layer for growth. To outcompete the microbiota, *Salmonella* must utilize nutrients generated because of the host inflammatory response [71]. *Salmonella* uses T3SS-1- and T3SS-2-mediated intestinal inflammation to engineer a nutrient niche characterized by increased availability of monosaccharides, amino acids, and respiratory electron acceptors [72].

During gastroenteritis (Figure 1), neutrophils transmigrate into the intestinal lumen in large numbers, giving rise to an abundance of fecal leukocytes, characteristic of inflammatory diarrhea. Neutrophils generate ROS that oxidize thiosulfate (S_4_O_3_^2−^) into tetrathionate (S_4_O_6_^2−^). The *ttrBCA ttrRS* gene cluster codes for tetrathionate reductases, enabling *Salmonella* to use tetrathionate as a terminal electron acceptor [73]. Through this mechanism, inflammation provides a respiratory electron acceptor that allows *Salmonella* to use anaerobic respiration instead of fermentation to produce energy for growth [74]. *Salmonella* can use tetrathionate as an electron receptor in anaerobic respiration, which gives them an advantage in the intestinal environment. Tetrathionate respiration enables *Salmonella* to utilize fermentation end products that the fermenting microbiota cannot consume. Using tetrathionate respiration for energy production presents *S*. Typhimurium with a vital growth advantage over competing microbes that rely on fermentation. Inactivation of genes required for tetrathionate respiration removes the ability of *S*. Typhimurium to outgrow the microbiota during intestinal inflammation [74]. These data indicate that tetrathionate respiration is one of the primary mechanisms enhancing the outgrowth of *Salmonella* in the inflamed gut.

The pathogenic strategy of *Salmonella* associated with gastroenteritis is to use virulence factors (T3SS-1, T3SS-2, and others) to elicit acute intestinal inflammation. This host response provides a new respiratory electron acceptor in the gut, enabling the pathogen to outgrow the microbiota in the lumen, thereby enhancing its transmission to the next host by fecal shedding of the organism (Figure 1). *Salmonella* thus uses the host to provide them with a substance that allows them to outgrow their competition [75].

*Salmonella* has evolved ways to subvert, mimic, antagonize, and exploit the defense strategy of vertebrate hosts with their virulence factors creating a novel niche that favors the growth of *Salmonella* to outcompete the resident microbiota [76,77]. *Salmonella* residing in the tissue face death by the host’s innate immune system, but acute inflammation changes the environment of the gut lumen to favor *Salmonella* growth (Figure 1) [77]. Luminal outgrowth is required to increase their abundance in intestinal contents during gastroenteritis for successful transmission to the next naïve host by the fecal–oral route. Diarrheal disease (gastroenteritis) flushes the intestinal lumen, removing the intestinal contents and the Salmonellae [6,77].
Figure 1*Salmonella*’s pathogenic strategy is to invade and trigger gastroenteritis. The purpose of the invasion is to trigger the host’s immune system, and the acute inflammation changes the environment of the intestinal lumen. The inflamed gut provides nutrients and a terminal electron acceptor for anaerobic respiration, which allows the Salmonellae to outcompete the fermenting microbiota [70,71,72,74,75,77].
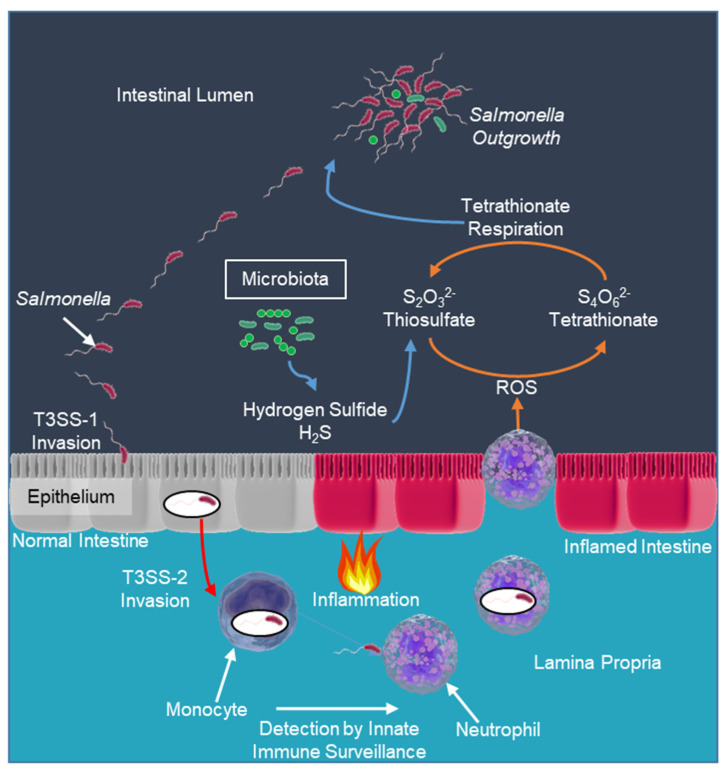



## 4. Salmonella Epidemiology

### 4.1. Salmonella in People

*Salmonella* infections affect roughly a million people in the USA each year. The CDC estimates that *Salmonella* infections can range from 645,000 to 1.7 million cases yearly, but only approximately 42,000 cases are laboratory-confirmed and reported to the surveillance system [78]. CDC Foodnet Annual Report for 2021 has *Salmonella* ranked second among laboratory-confirmed bacterial foodborne pathogens but the highest in hospitalizations, deaths, and outbreak-associated infections [34]. The United States Department of Agriculture (USDA) Economic Research Service reports that in 2013, *Salmonella* was responsible for approximately 11% of all foodborne illnesses, second only to Norovirus infections. Non-typhoidal *Salmonella* is estimated as the leading cause of hospitalizations (35% or 19,000 cases) and deaths (28% or 378 deaths) caused by foodborne illnesses linked to a specific pathogen. *Salmonella* is ranked first among 15 pathogens in terms of economic burden, estimated at $3.7 billion in a typical year. Ninety percent of the burden is due to deaths ($3.3 billion), eight percent due to hospitalization ($294 million), and the remaining two percent to non-hospitalized cases. The economic burden can range from $193 million to $9.5 billion annually [1].

The most recent CDC LEDS *Salmonella* Annual Report (2016) reported there were 46,623 laboratory-confirmed human *Salmonella* infections. Approximately 22% of these laboratory-confirmed *Salmonella* infections were in children four years of age or younger. The most frequently reported serovars were *Sal.* Enteritidis (16.8%), *Sal.* Newport (10.1%), *Sal.* Typhimurium (9.8%), *Sal.* Javiana (5.8%), *Sal. enterica* serovar I 4,[5],12:i:- (4.7%), *Sal.* Infantis (2.7%), *Sal.* Muenchen (2.6%), *Sal.* Montevideo (2.2%), *Sal.* Braenderup (2.1%), and *Sal.* Thompson (1.7%) [2]. The US Department of Health and Human Services’ “Healthy People 2030” made it a national health objective to reduce *Salmonella* infections by 25% by 2030. The 2030 objective target is an incidence rate of 11.5 per 10^6^ people; in 2021, the incidence rate was reported as 13.3 per 10^6^ people, which is a decrease from the 2016–2018 report [79].

### 4.2. Environmental Salmonella in Animal Facilities

The incidence of *Salmonella* has been extensively studied in animals and environmental sites. There is a wide array of environmental niches in which *Salmonella* can survive. *Salmonella* may be disseminated in various water sources such as effluent discharges, agricultural runoff, excretions by wild animals, and freshwater. Sediments may protect enteric organisms like *Salmonella* from stresses in aquatic environments and provide some nutrients. Water contaminated with animal waste has the potential to proliferate and disseminate *Salmonella* by wild animals [80].

Farm environments can easily be affected if there are outbreaks of *Salmonella* among animals or if some of the animals on the farm are asymptomatic carriers. Other than animal-to-animal transmission, additional factors for the on-farm environmental spread of *Salmonella* include recycling of (manure) lagoon wastewater for flushing, contaminated feeds, inadequately controlled rodent and wild bird populations, contaminated rendering trucks being driven into animal areas and use of the same loader for transporting dead animals and moving feeds without appropriate cleaning and decontamination [80]. Movement of animals can also lead to the spread of *Salmonella* by introducing an infected or carrier animal into a herd of non-infected animals [81]. Due to the stress of being moved or transported between premises, animals are at risk of being more susceptible to infection [82]. Many production animals have subclinical infections that lead to widespread environmental contamination. This makes internal and external biosecurity measures critical to restrict *Salmonella* movement within a farm [83]. Some of the best practices for the reduction in environmental *Salmonella* are worker education in biosafety and cleaning practices [84,85,86,87,88,89]. Sources for contamination of environmental sites are so diverse that absolute elimination of *Salmonella* in the outdoor farm environment is impossible. However, addressing efforts to prevent introduction, minimize pathogen load, and prevent unintended distribution spread may assist in reductions into the food chain [80].

### 4.3. Salmonella in Animals

The host–host transmission of *Salmonella* is primarily by the fecal–oral route. After a host becomes infected, most of the time, the host will resolve the salmonellosis, and shedding will stop. Still, a few infected individuals will become carriers and will intermittently shed *Salmonella* in their feces for long periods. These animals can act as reservoirs for the pathogen. Meat-producing animals can be sources of food contamination for humans by fecal contamination of vegetables, fruit, and nuts or from fecal contamination of carcasses upon slaughter [90]. It is also possible that wildlife can serve as reservoir hosts by being asymptomatic carriers, causing sporadic cases of salmonellosis by contamination of feeding places [91]. It has been observed that supershedders can lead to persistent shedding of *Salmonella*. However, in low-shedding animals, constant reinfection and host-to-host transmission lead to persistent *Salmonella* shedding, and persistent shedding can be interrupted by breaking the cycle of reinfection [92].

*Salmonella* can be found in many different domestic and wild animals [47]. Poultry, swine, cattle, wild birds, rodents, pets, and exotic animals can all be reservoirs for *Salmonella*. Animals such as companion animals (dogs and cats) and exotic animals (reptiles, birds, and amphibians) can pose a risk of infecting humans or other animals in the environment [47]. A study looking at bacterial species associated with hospital-acquired infections (HAIs) (both human and veterinary HAIs) found *Salmonella* as the fourth highest (15% of reported HAIs) bacterial pathogen reported [93]. *Salmonella* is insidious, with animals being asymptomatic shedders where they can shed the bacterium in high numbers. This poses a risk for nosocomial infections and zoonotic infections to the personnel working close to these animals [93].

*Salmonella* can exhibit different symptoms in animals. Common symptoms for cattle are diarrhea, fever, and dehydration. Cattle can suffer from abortions and subclinical symptoms like reduced milk production [94]. Clinical salmonellosis in pigs is usually enterocolitis or septicemia, as with *Sal*. Cholerasuis infections [95]. Horses usually exhibit colic from salmonellosis [96]. Chickens typically are asymptomatic with most *Salmonella* serovars except for *Sal*. Gallinarium and *Sal*. Pullorum—both have high levels of mortality; chickens infected with *Sal*. Gallinarium usually appear normal 24–36 h prior to death. With all other serovars, infections are asymptomatic. Chickens pose the highest risk to public health because domestic fowl constitute one of the largest reservoirs of *Salmonella* [97]. An additional concern with chickens is that *Salmonella* serovars such as *Sal*. Enteritidis can be vertically transmitted from hen to eggs by transovarian infection in the laying hen [98].

Several animal models have been used to study *Salmonella*. Mice, rabbits, zebrafish, rats, cattle, chickens, and rhesus macaques have all been used to study the disease of *Salmonella* [99,100,101,102,103,104,105]. Mice are used for studies of *Salmonella*, but studies involving *Sal*. Typhimurium tends to be a model representing typhoid fever in humans. This is because of the *Sal*. Typhimurium in mice has symptoms very similar to *Sal*. Typhi in humans [104]. The calf animal model tends to be a very good model for *Salmonella*-induced enteritis except for *Sal*. Dublin [102,104]. *Salmonella* and phage interventions in animal models have included mice, chickens, and calves [106,107,108,109,110,111].

### 4.4. Salmonella Outbreaks in Veterinary Hospitals

*Salmonella* outbreaks have repeatedly been shown to be a constant risk to all types of veterinary hospitals. The results have been costly, both financially and in significant morbidity and mortality among patients and zoonotic infections among hospital personnel [88,112,113,114,115,116]. Most of these nosocomial outbreaks involved horse patients, which is not surprising because horses exhibit severe symptoms such as colic, while other animals, like cattle, can be asymptomatic [94,96]. The most recent nosocomial outbreak was reported in 2014 and was discovered by retrospective analysis that concluded the outbreak lasted from 1 January 2006 to 1 June 2011 [88].

*Sal*. Infantis, *Sal*. Newport (two outbreaks), *Sal*. Oranienberg, and *Sal*. Typhimurium (six outbreaks) were the serovars responsible for the more recently reported *Salmonella* nosocomial infections involving veterinary hospitals [88,112,113,114,115,116,117]. Veterinary hospital *Salmonella* outbreaks all have the universal feature of widespread environmental contamination [118,119,120]. Animals are likely the initial source of environmental contamination, which leads to the environmental contamination becoming the source of continued infection in new patients [113,119]. Biosecurity is critical to managing the transmission of *Salmonella* in veterinary settings [93]. For this reason, people play a crucial role in the transmission of HAIs. Education programs for personnel in hygiene, proper use of personal protection equipment, movement control, cleanliness of equipment, managing high-risk groups, and the benefits of continued compliance are the best practices to reduce transmission to a baseline or acceptable endemic level [93,113,118,120,121,122].

### 4.5. Antimicrobial-Resistance Salmonella

Non-typhoidal salmonellosis is usually a self-limiting illness, but antimicrobial treatment is recommended for patients with severe infections or those at risk for complications. Antimicrobial-resistant *Salmonella* infections may be more severe and have increased rates of hospitalizations [123]. There has been an increase in the incidence of antimicrobial resistance in *Salmonella* found in humans, with a 40% increase in the annual number of culture-confirmed infections with clinically important resistance [124]. The use of antimicrobials in animals can promote the emergence and spread of resistant bacteria and become a threat to human health [125]. Antimicrobial use in animals represents 73% of all antimicrobials used worldwide, which is a contributing factor to increased antimicrobial resistance [126]. This means there needs to be a One Health approach to detecting and controlling antimicrobial resistance [124]. Studies have shown that non-typhoidal serotypes isolated from non-human origins possess more antimicrobial-resistant genes, virulence genes, mobile genetic elements, and antimicrobial-resistant phenotypes than those serovars from human origins [127]. Antimicrobial resistance is a concern in humans and animals, but interventions in combating antimicrobial resistance in *Salmonella* need to be focused on animals.

## 5. Future Trends in Approaches to Combat *Salmonella*

*Salmonella*’s being a pathogen of concern and its prevalence have led to new novel interventions to combat this organism. There are many different approaches being investigated, but this review will cover two novel approaches. The first novel approach is exploring bacteriophage and concerns with bacteriophage resistance. The second novel approach is the use of computational approaches such as machine learning algorithms to determine factors associated with the prevalence of *Salmonella* and next-generation sequencing to track *Salmonella*. The use of these computational approaches can assist with a better understanding of *Salmonella* that allows the implementation of interventions to reduce *Salmonella.*

### 5.1. Bacteriophage

#### 5.1.1. Introduction and Phage Therapy

The French-Canadian microbiologist Félix D’Hérelle devised the term “bacteriophage”, which means bacteria-eater [128]. Phages are viruses; like all other viruses, they are obligate intracellular parasites of cellular organisms with their life cycle within a host cell. The basic life cycle involves using bacterial cellular metabolism to produce new phage particles, release them from their cellular confines, and infect new cells [129].

This cycle of infection, replication, and release of bacterial cells gives phages the opportunity to be used as highly specific antimicrobial agents [130]. This final step of the lytic life cycle, in which phages kill the bacterial cells, is the cornerstone of the idea of using phages as an antimicrobial agent [131]. This replication cycle is why the term “self-replicating pharmaceuticals” was coined for phage (Table 1) [132].

Antibiotics have been the main treatment for bacterial diseases, but antibiotic resistance has been a growing concern. It is estimated that by 2050, ten million people a year will die from multidrug-resistant (MDR) bacterial pathogens. This has renewed interest in phage and phage therapy [133]. Thirty-five case reports of the emergency use of phage therapy were reported from 2008 to 2021. Studies have been limited on the safety and toxicity of phage therapy [134]. Phage is widely regarded as safe, with the Federal Bureau of Drugs and Administration (FDA) approving phage preparations as “Generally Recognized as Safe” (GRAS) in food preparations (Table 1) [135,136,137].

Phages can be very effective in reaching the body by many different routes. Phages can spread throughout the body and even cross the blood–brain barrier (Table 1) [136,138]. When treating with phages, where the bacterium is found, so can the phage be found. The proposed mechanism is called the “trojan horse” mechanism, by which the phage-infected bacterium moves through the body and thus carries the phage with them [136]. Phages have a narrow specificity to their host bacteria and will not disrupt the body’s microbiota; they increase in number after administration due to replicating in host cells; and they can have lytic activity against MDR bacterial pathogens, and can penetrate biofilms [133,134]. Phages have also been proposed as a treatment in conjunction with antibiotics. Phages have been shown to re-sensitize MDR bacterial pathogens to antibiotics. This phage–antibiotic synergy could combat many pathogens [139,140].

No harmful effects have been observed with phage treatment, but the purity of phages has been a problem. Safety issues involve crude phage preparations due to LPS, peptidoglycan, and additional inflammatory components from lysed bacteria (Table 1). This can be alleviated with phage purification by density gradient purification or column chromatography [138,141,142]. A potential downside of phage therapy is that the phage undergoes an alternative lifecycle called the lysogenic cycle, in which the phage DNA integrates into the host’s DNA [133]. This lysogenic conversion transfers undesirable genetic material, such as virulence factors or antibiotic resistance (Table 1) [133,135,136]. Another major concern with the use of phages is the emergence of phage-resistant variants. To combat the problem of phage resistance, it has been proposed to use multiple phages or a “phage cocktail” (Table 1). The phage cocktail could prevent the emergence of phage resistance [143]. A study by Dalmasso et al. (2016) found that a cocktail of three phages inhibited *Escherichia coli*’s growth while preventing phage-resistant mutants’ emergence [144]. Phage resistance should not be underestimated, and understanding phage resistance costs or benefits to a host bacterium is critical in the progression of phage therapy (Table 1) [145].
microorganisms-12-02249-t001_Table 1Table 1The pros and cons of phage and phage therapy.Pros of Bacteriophage TherapyCons of Bacteriophage TherapyPhages can be used to target multidrug-resistant bacterial pathogens [133].Phages have a narrow host range and would be ineffective against non-targeted bacteria or even different strains of the same targeted bacteria [134].Phages can penetrate biofilms [134]Crude phage lysates or preparations can be toxic due to LPS, peptidoglycan, and other inflammatory components of lysed bacteria [138,141].Phage preparations have been approved by the FDA as “Generally Recognized as Safe” (GRAS) in food preparations [135,136,137].Phages may harbor virulence genes or other undesirable genetic material and can confer these to a bacterium called lysogenic conversion where the phage undergoes an alternative lifecycle in which the phage DNA integrates into the host’s DNA [133].Where a bacterial host goes in the body, so goes the phage. Phages can even cross the blood–brain barrier [136,138].Widespread use of phages would lead to phage-resistant strains of bacteria [146].A self-replicating pharmaceutical due to the nature of phage infecting, replicating or amplifying, and lysing the host bacterial cell [132].
Phage-resistant bacterial organisms selected for by use of the phage may have reduced virulence or be attenuated when compared to phage-sensitive bacteria [147].



#### 5.1.2. Bacteriophage Resistance

Bacteria and phages are in a permanent arms race with co-evolution driving mechanisms to evade and the latter adapting and avoiding evolved defense systems [146]. There are several steps of phage infection in a host bacterium: the phage attaches to the surface of the cell, injects the phage genome into the bacterial cell, assembles phage proteins, and releases progeny phages. At any of these stages, phage infection can be inhibited or aborted [147,148]. The most common mechanism of phage resistance is to prevent phage adsorption.

Phage resistance by preventing phage adsorption occurs by the modification of surface phage receptors, hindering access to phage receptors, or by producing competitive inhibitors [145]. Phage can use many different surface molecules as phage receptors; this includes outer membrane proteins, flagella, pili, capsule, and teichoic acids in Gram-positive bacteria and LPS in Gram-negative bacteria [147,149]. The O-antigen region of LPS is also a receptor for numerous phages [147]. Modifications to these structures can lead to phage resistance if any of these are the phage’s receptor [150]. Outer membrane vehicles can also be a mechanism by being a non-replicating nanostructure composed of a membrane and membrane structures produced during bacterial growth. These molecules can act as decoys for phages to bind, thus reducing phage titers (phage numbers) [148]. Bacteria can produce extracellular matrixes that are a physical barrier to phages, or masking proteins that block phage adhesion, and even flagellar phase variation as seen in *Salmonella* can be a mechanism to prevent phage attachment and adsorption [146,148,149,150].

The mechanism by which bacteria can block phage DNA injection is superinfection exclusion (Sies) systems. These systems are proteins that block the entry of phage DNA into the host bacterium. Many of these systems are encoded by prophage or lysogenic phage genes, and they protect phages by preventing other phages from infecting the same cell [148,150].

If phage DNA is injected, an innate defense system can cleave injected DNA. These systems include the restriction–modification system, the defense island system associated with restriction–modification, and prokaryote argonaute proteins. Adaptive immune systems include the CRISPR-Cas system, where foreign DNA is inserted into the CRISPR loci, and when CRISPR ribonucleic acids (RNAs) bind to a complementary nucleic acid that has entered the cell, the DNA is degraded by nucleases [145,148,150,151].

Phage-inducible chromosomal islands (PICI) can detect infection by phages. The PICI will be excised, circularized, replicated, and packaged. These PICI are packaged into the capsid of the phage particle instead of the phage genome. The cell will lyse, but the particles released will carry the PICIs instead of the phage genome. These will infect other cells, and the PICI will incorporate itself into the new cell [148,152].

The final mechanism of phage resistance is abortive infection (Abi) systems. These systems are not fully understood, but it is known that these systems cause bacterial cells to die [142,148,149,150]. These systems can be triggered at many different stages of the infection cycle [148]. These altruistic bacterial systems trigger the cell to commit suicide, in which the surrounding bacterial population is protected by the phage being contained within the dead cell [153,154,155].

Many of these mechanisms can affect the virulence of the bacterium. These modifications can lower the fitness compared to non-resistant strains [147]. There have been many different studies that have concluded that phage resistance leads to decreased virulence in *Listeria monocytogenes*, *Bacillus thuringiensis*, *Vibrio cholerae*, *Dickeya solani*, *Flavobacterium columnare*, *Staphylococcus aureus*, *Serratia marcescens*, *Shigella flexneri*, *Klebsiella pneumoniae*, and *Salmonella* serovars [156,157,158,159,160,161,162,163,164,165,166]. Capparelli et al. (2010) found that *Salmonella* resistance to phages was due to the lack of the O-antigen on the LPS, which conferred attenuation in mice [107]. It appears that phage resistance can help the bacterium survive viral infection, but the trade-off is a fitness cost that typically affects the virulence of the bacterium in a host organism.

### 5.2. Computational Analysis in Epidemiology

#### 5.2.1. Machine Learning Algorithms

To quote Leonard M Schuman, “Any science is as objective as its capability of measuring the events which it purports to be observing and relating. Epidemiology has not been exempt from the usual evolutionary development of this necessary aspect of its methodology.” [167]. Abraham Lilienfeld (1980) wrote a review titled “Advances in Quantitative Methods in Epidemiology”, in which he discussed that in the 1950s, the new statistical tool was the 2 × 2 contingency table and estimates of relative risk and odds ratios. Lilienfeld (1980) discussed the development of a “logistic regression” method that uses many factors that might influence the occurrence of a disease to calculate estimates of relative risk and tests of significance. Lilienfeld (1980) stated that the “entry of the computer” into data analysis makes a “logistic regression” possible, and the “current” problem is epidemiologists’ unfamiliarity with using computers and the need for training on computer usage [167]. A publication in 1997 discussed the usefulness of a computer program to calculate 2 × 2 contingency table data as the program is an easy and quick “epidemiological calculator” [168]. Zocchetti et al. (1997) discussed using “some algebra” for 2 × 2 contingency table calculations for prevalence risk ratios versus the not-so-easy prevalence risk ratios calculations with advanced statistical tools like logistic regression [169]. Even an article published in 2017 discusses the usefulness of manually calculating odds ratios and relative risk with 2 × 2 contingency tables [170]. Lilienfeld hoped that the introduction of computers would bring the transition from more classical epidemiological methods to more advanced methods, but epidemiologists can be stubborn with the adherence to simple analytical methods with the concern that readers might not understand more complex methods [171,172].

Novel computational modeling strategies are being utilized in scientific literature. One area of interest is in “machine learning” algorithms, and these algorithms could be beneficial to epidemiologists [173]. These models “learn” from the data to improve their performance, and the analysis allows for the identification of “important” variables [174]. Machine learning shines within descriptive epidemiology, which is the field that describes associations between multiple variables and identifies patterns within the data [174,175]. Machine learning algorithms have three methods of learning: supervised, unsupervised, and semi-supervised. Supervised learning is when the outcome is known for each observation. Unsupervised learning attempts to identify relationships and groups within the data, but the outcomes are not known. Semi-supervised learning is a mixture of supervised and unsupervised where some outcomes are known and others might have missing data [176,177]. Data are usually split into two groups: training data and test data. Training data are a randomly selected subset of data that is used to train the machine learning algorithm. The test dataset is used to evaluate the performance of the model to predict the outcome [173,175,177,178]

Markov Chain Monte Carlo (MCMC) analysis is a method used to simulate parameter distributions of interest, such as generalized linear model parameters. It is particularly useful for handling difficult types of analyses and is commonly used for Bayesian analysis. MCMC methods involve repeatedly querying datasets to determine the probability distribution function of quantities of interest. The resulting sequence of values forms a Markov chain that can be analyzed to find best-fit values and confidence intervals. MCMC approaches can provide advantages over methods based on standard maximum-likelihood estimation (MLE) and allow for the simultaneous estimation of parameters for complex models [178]. MCMC in a Bayesian framework allows the posterior probability distribution to be approximated computationally, which revolutionizes infectious disease modeling [179].

Another type of machine learning algorithm is a decision tree model. A decision tree is a model that separates data into smaller and smaller partitions until each observation is classified according to the outcome of interest [177]. Random Forest analysis is a machine learning technique that combines multiple decision trees to make predictions for the outcome [177,180]. It is a useful tool for epidemiologists as it allows for the interpretation of complex association patterns in epidemiological data. Using random forests, researchers can identify relevant features and understand their relationships with the outcome of interest. Combining random forests with Bayesian network surrogate models further enhances interpretability by providing a deeper understanding of the association patterns [180]. The random forest model provides predictive accuracy, but they are considered black-box models. This is because it is difficult to retrace how the model came to a specific prediction [181].

Epidemiologists should transition to machine learning because it offers new tools to tackle problems for which classical methods are not well suited [182]. Machine learning can be used for descriptive epidemiology to identify important associations and predictors of outcomes [174]. Integrating machine learning algorithms with existing methods can improve the understanding of health and disease [173]. However, language and technical barriers between the fields of epidemiology and machine learning need to be addressed [178]. Epidemiologists can benefit from learning the concepts and terminology used in machine learning literature [178]. By incorporating machine learning into epidemiologic research, there are opportunities to enhance the field and improve the safety and efficacy of applications [183].

#### 5.2.2. Whole-Genome Sequencing in Bacterial Epidemiology

During the 2000s, next-generation sequencing greatly increased sequencing capabilities, allowing the ability to sequence the whole genome of microorganisms [184]. Whole-genome sequencing (WGS) technology has revolutionized the field of bacterial epidemiology by providing valuable insights into the epidemiology and pathogenesis of infectious diseases. WGS has become more affordable and accessible for microbiological laboratories, allowing for an improved understanding of disease ecology and control strategies [185]. WGS has significantly increased the amount of information available for studying infectious diseases and has improved the precision of epidemiological inferences. The use of WGS in the surveillance of bacterial pathogens has proven effective in outbreak investigations, source tracking, and variant analysis [186,187]. Whole-genome sequencing (WGS) technology is increasingly used for the epidemiology of *Salmonella*. WGS allows for the analysis of *Salmonella* isolates obtained from various sources, such as food products, animals, and humans, to better understand their genetic characteristics and relationships [185,186].

Short-read sequencing was the next advancement in sequencing technology after first-generation traditional Sanger sequencing. Short-read technology is the sequencing of short (250–800 base pairs (bp)), clonally amplified DNA molecules sequenced in parallel [188]. Illumina sequencing platform (36–300 bp) is based on the “sequencing by synthesis” (SBS) approach that involves DNA-polymerase-dependent nucleotide incorporation on the extended DNA chain [188,189]. Illumina’s technology is based on the SBS with a fluorescent-labeled reversible terminator technology. This, along with paired-end sequencing, makes it the most accurate base-by-base sequencing technology, with an error rate of 0.1% [188].

Long-read sequencing technology (third-generation sequencing technology) can generate sequences greater than 10,000 bp directly from native DNA. Inaccuracies plagued early iterations, but recent modifications have improved the accuracy. The two primary long-read technologies are Pacific Biosciences (PacBio) and Oxford Nanopore Technology (ONT) [188]. The two technologies operate on different principles. PacBio can be used for whole-genome de novo assembly due to the read length being up to 300 kb, but it has the disadvantage of the high cost associated with sequencing. ONT reads average from 10 to 30 kb, but they have a very high error rate [188,189]. ONT long-read technology is based on a nanopore technology where single-stranded DNA is passed through a biological pore, and the electrical current is measured as each base is passed through [188,189]. Deep learning algorithms are used to translate the electrical signals into a DNA sequence [190].

Short-read sequencing technology results in incomplete genome assemblies only considered draft genomes. Long-read sequencing technology is highly error-prone. The new frontier in genome assembly is to make a hybrid assembly based on short-read and long-read sequencing [191]. With hybrid assembly, the genome is assembled first with the long reads, and then the short reads are used to polish the long-read assembly [192]. This provides a more polished complete reference genome because the short reads are of higher quality than the long reads, while the long reads provide a backbone for the short reads and fill in gaps [193].

Short-read sequencing technology’s incomplete draft genomes have been commonly used to identify and characterize *Salmonella* bacteria [194,195]. Serotyping, detecting antimicrobial resistance and virulence genes, plasmid detection, and phylogenetics can be performed with short reads. Phylogeny approaches are core-genome MLST (cgMLST) (aligns core genes), whole-genome MLST (wgMLST) (aligns core and accessory genes), single-nucleotide polymorphism (SNP) (core SNP alignment and SNP matrix), and K-mer based methods (pair-wise comparison of nucleotide blocks) [196]. SNP phylogeny or SNP detection first requires a closely related genome reference, in which the choice of reference is critical. Draft genomes contain too many contigs and are poor at calling accurate SNP positions. A reference that is too distant will provide fewer reference positions, and fewer SNPs will be discovered [196]. Reads are aligned to a reference, and variants are detected [188]. Variants are SNPs, insertions, deletions, and structural variants such as duplications, inversions, and translocations [191,197]. Tools available for SNP calling are SAMtools, GATK, and Freebayes. Popular specialized pipelines for SNP calling for bacterial genomes are Snippy, CFSAN SNP Pipeline, NASP, and BactSNP [187,194,196,198].

## 6. Conclusions

In conclusion, there are a great number of publications about *Salmonella*. Many researchers have extensively studied *Salmonella* as a pathogen. The use of next-generation sequencing technologies in strain identification and source tracking is revolutionizing the ability of public health officials to respond to outbreaks quickly. However, implementation of artificial intelligence technologies such as machine learning algorithms/models in *Salmonella* outbreak investigation for identification of source-specific factors or factors contributing to prevalence is in its infancy as publications on this topic of *Salmonella* are lacking. Adoption of more machine learning technologies in surveillance studies of *Salmonella* can help us better understand the dynamics of *Salmonella*. Machine learning technologies and the integration of next-generation sequencing will transform the field of epidemiology and specifically help with the control of *Salmonella* into the food production chain by identifying areas to target for interventions. Short-read sequencing remains essential for generating detailed draft genomes, while long-read sequencing provides opportunities for more comprehensive genome assemblies. Machine learning algorithms further enhance our ability to analyze complex data and identify important associations within epidemiological datasets. A key area of emerging interest is the role of bacteriophages in *Salmonella* epidemiology. Phages interact with *Salmonella* in ways that influence bacterial evolution, population dynamics, and pathogen control strategies. Incorporating phage-related research into *Salmonella* genomics provides a deeper understanding of how phages impact bacterial behavior and resistance patterns. Many computational and bioinformatics tools were not developed specifically for *Salmonella* but can be utilized for *Salmonella* research. To expand and further develop the implementation of these computational and bioinformatic tools, more research needs to be conducted, including research on *Salmonella*. Additional research in machine learning algorithms specifically will improve the field of epidemiology and lead to more accurate predictions of sources, the best choice of appropriate algorithms, and/or provide investigators with the knowledge of what well-defined inputs are necessary for high-quality training data for models. The combination of a better understanding and implementation of artificial intelligence, such as machine learning algorithms, and continued improvements in sequencing technologies has the potential to provide information to public health officials to develop containment strategies or even predict/forecast future outbreaks. Advancements in sequencing technologies and computational methods, combined with a focus on phage interactions, will likely continue to drive significant progress in the study of *Salmonella*, ultimately leading to more effective strategies for disease prevention and control.

## Data Availability

No new data were created or analyzed in this study. Data sharing is not applicable to this article.

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
