# Peer review of "Bridging Classical Methodologies in Salmonella Investigation with Modern Technologies: A Comprehensive Review"

_microorganisms, 2024, doi:10.3390/microorganisms12112249_

Round 1
Reviewer 1 Report
Comments and Suggestions for Authors
The manuscript entitled “Unraveling Salmonella Epidemiology Through Genomics, Phage Research, and Machine Learning: A Comprehensive Review” highlights the integration of these modern methodologies in advancing our understanding of Salmonella and underscores the role of phages in shaping bacterial populations and influencing disease dynamics. There are many simple descriptions in the main text, and there is a lack of in-depth review.
Several major findings are either repetitive of previous studies or questionable by available data and analyses.
Some of the content is verbose and needs to be refined
I think authors should better state novelties of the present paper with respect to existing literature.
When the abbreviation first appears, please using the full name meanwhile.
Line 46: Recent literature suggests additional subspecies doi.org/10.1016/j.ygeno.2021.07.003.
Line 80: Reference about MLST should be cited here.
Line 293- 296. S. Typhimurium was the most dominant serovar causing human infection in China. (The temporal dynamics of antimicrobial-resistant Salmonella enterica and predominant serovars in China; Y. Wang, Y. Liu, N. Lyu, Z. Li, S. Ma, D. Cao, et al. Natl Sci Rev 2023;10: nwac269. Accession Number: 37035020 PMCID: PMC10076184 DOI: 10.1093/nsr/nwac269)
Line: 340: Genomic analysis of almost 8,000 Salmonella genomes reveals drivers and landscape of antimicrobial resistance in China; Y. Wang, X. Xu, B. Zhu, N. Lyu, Y. Liu, S. Ma, et al.; Microbiol Spectr 2023; 11: e0208023. Accession Number: 37787535 PMCID: PMC10714754 DOI: 10.1128/spectrum.02080-23.
Worldwide Epidemiology of Salmonella Serovars in Animal-Based Foods: a Meta-analysis. R. G. Ferrari, D. K. A. Rosario, A. Cunha-Neto, S. B. Mano, E. E. S. Figueiredo and C. A. Conte-Junior Appl Environ Microbiol 2019; 85 Issue 14. Accession Number: 31053586 PMCID: PMC6606869 DOI: 10.1128/AEM.00591-19.
Comments on the Quality of English Language
moderate revision
Reviewer 2 Report
Comments and Suggestions for Authors
Salmonellosis still represents a significant problem in public health, both in non-developed and developed countries.
The manuscript is very well written, scientifically based and detailed, it shows the historical development of salmonellosis diagnostics, from the discovery of the bacterium to the most modern approach in diagnostics and therapy.
Numerous references, significant for this topic, are cited. The manuscript is a very good review article for anyone interested in salmonellosis, from diagnostic to clinical and therapeutic approaches.
The manuscript is clearly written and easy to read. Authors have described the topic in detail, from the earliest do the most modern epidemiological methods.
It was a great pleasure to read it and, in my opinion, the manuscript will be very interesting to readers. It think it may be publish in the present form.
Reviewer 3 Report
Comments and Suggestions for Authors
Please, find general comments and specific remarks in the attached file

Round 2
Reviewer 3 Report
Comments and Suggestions for Authors
The authors have taken into account most of the comments and the manuscript has been improved accordingly. However, the conclusion needs to be revised again, the statement that publications regarding current sequencing technologies is laking for Salmonella is not true, there is a hudge of publications when seraching Salmonella and NGS, but it should be true with machine learning. I think the authors must istinguish NGS with machine learning. The first sentence of the conclusion is confusing beacause we don't know what is mean under "these current technologies..."
